# High-resolution greenhouse gas flux inversions using a machine learning surrogate model for atmospheric transport

Nikhil Dadheech[1,*], Tai-Long He[1,*], and Alexander J. Turner[1]

[1]Department of Atmospheric and Climate Science, University of Washington, Seattle, WA, USA
[*]These authors contributed equally to this work

**Correspondence:** Alexander J. Turner (turneraj@uw.edu)

**Abstract.** Quantifying greenhouse gas (GHG) emissions is critically important for projecting future climate and assessing the impact of environmental policy. Estimating GHG emissions using atmospheric observations is typically done using source-receptor relationships (i.e., "footprints"). Constructing these footprints can be computationally expensive and is rapidly becoming a computational bottleneck for studying GHG fluxes at high spatio-temporal resolution using dense observations. Here we demonstrate a computationally efficient GHG flux inversion framework using a machine learning emulator for atmospheric transport (FootNet) as a surrogate for the full-physics model. The footprints generated by FootNet are at approximately one-kilometer resolution. We update the architecture of the deep-learning model to improve the performance in a GHG flux inversion. We find that the posterior fluxes estimated with FootNet footprints are in good agreement with the posterior fluxes estimated with STILT footprints. We observe that the more simplistic representation of transport in the machine learning model helps to mitigate transport errors. This flux inversion using a machine learning surrogate model only requires meteorological data, GHG measurements, and prior fluxes. Constructing footprints using FootNet is $650\times$ faster than the full-physics atmospheric transport model on similar hardware. This speedup allows for the computation of footprints "on-the-fly" during the GHG flux inversion (i.e., computed as needed, rather than archiving for future use) and makes near-real-time emission monitoring computationally possible. This work alleviates a major computational bottleneck with inferring GHG fluxes with next generation dense observing systems.

## 1 Introduction

Carbon dioxide ($CO_2$) and methane are the two most powerful greenhouse gases (GHGs). Together, they account for more than 85% of the total GHG radiative forcing since pre-industrial times (IPCC, 2023). As such, it is important to quantify the GHG sources and sinks in order to project future climate. Near real-time quantification of GHG emissions is the key to identifying the intermittent super-emitters which often dominate the emission budget. However, the large computational and storage costs associated with full physics atmospheric transport models in the current inversion framework limit our ability to perform near-real-time emissions monitoring from urban to global scales (Roten et al., 2021; Varon et al., 2022a; Cartwright et al., 2023; Fillola et al., 2023; Nayagam et al., 2023; Steiner et al., 2024; Janardanan et al., 2024). Here, we use FootNet (He et al., in press), a computationally efficient deep-learning model, to emulate a full-physics atmospheric transport model and

conduct GHG flux inversions. This work shows the feasibility of using a machine learning (ML) emulator for near-real-time computation of source-receptor relationships and infer hourly GHG emission fluxes at the kilometer scale from atmospheric observations.

Previous work has shown the importance of point sources for methane emissions (e.g., Brandt et al., 2014; Zavala-Araiza et al., 2015; Frankenberg et al., 2016; Duren et al., 2019; Lauvaux et al., 2022; Chen et al., 2022; Cusworth et al., 2022; Sherwin et al., 2023; He et al., 2024), and urban & localized sources for $CO_2$ (e.g., Hutyra et al., 2014; Janardanan et al., 2016; Turner et al., 2020; Wu et al., 2020; Kiel et al., 2021). These sources represent a small geographical area and yet dominate the GHG emission budget. The emission fluxes from these "super emitters" are often observed to have a heavy-tailed distribution (Brandt et al., 2014; Zavala-Araiza et al., 2015; Frankenberg et al., 2016; Duren et al., 2019; Chen et al., 2022; Cusworth et al., 2022). In other words, a small number of point sources are responsible for a large fraction of the total emission budget despite representing a small fraction of the land area. As such, studying these point sources requires densely spaced measurements due to their localized nature. Fortunately, there has been a proliferation of next-generation dense observing systems for GHGs over the past decade including both space-borne instruments e.g. OCO-2 (O'Dell et al., 2012; Crisp et al., 2012; Eldering et al., 2012; Hammerling et al., 2012), OCO-3 (Eldering et al., 2019; Taylor et al., 2020), TROPOMI (Veefkind et al., 2012), and low-cost urban monitoring networks e.g. BEACO$_2$N (Shusterman et al., 2016).

Measurements from dense observing systems can be used to quantify surface fluxes. The inverse modelling of GHG emissions often requires one to know the upwind region of influence on the atmospheric measurements. This region of influence is also known as the "source-receptor-relationship" or the measurement "footprint" (cf. Rodgers, 2000). The $i^{\text{th}}$ observation ($y_i$) can be related to the $m$ surface fluxes using the associated footprint:

$$y_i = \boldsymbol{h}_i \boldsymbol{x} + b_i \tag{1}$$

where $\boldsymbol{h}_i$ is the $1 \times m$ footprint for the $i^{\text{th}}$ observation, $\boldsymbol{x}$ is an $m \times 1$ vector of surface fluxes, and $b_i$ is the background concentration for the observation. Here, $\boldsymbol{h}_i$ describes the sensitivity of observation $y_i$ to $m$ surface fluxes with units similar to ppm/($\mu$mol m$^{-2}$ s$^{-1}$). Similarly, the vector of the $n$ observations ($\boldsymbol{y}$; $n \times 1$) can be related to the surface fluxes as:

$$\boldsymbol{y} = \mathbf{H}\boldsymbol{x} + \boldsymbol{b} \tag{2}$$

where $\boldsymbol{b}$ is $n \times 1$ vector of background concentrations and $\mathbf{H}$ is an $n \times m$ Jacobian matrix representing the atmospheric transport such that the $i^{\text{th}}$ row of $\mathbf{H}$ describes the sensitivity of $i^{\text{th}}$ observation ($y_i$) to the $m$ surface fluxes.

Researchers construct this source-receptor relationship ($\mathbf{H}$) by running a Lagrangian model $n$ times or an Eulerian model $m$ times. The choice of construction of $\mathbf{H}$ will depend on the size of both $n$ and $m$. There are approximately 800 observations and 15M state vector elements in each inversion run for this study. Therefore, constructing $\mathbf{H}$ will require $\sim 800$ simulations in the Lagrangian framework or $\sim 15$M simulations in the Eulerian framework. The Eulerian models are not grid agnostic, and the corresponding computational expense increases as the spatial resolution increases (Steiner et al., 2024). For example, the Integrated Methane Inversion (IMI) is an Eulerian-based framework focusing on the regional scale but is limited to 25 km at present (Varon et al., 2022b). Variational methods such as 4D-var can be used with large state and observation space. However,

it requires computing an adjoint, which is a computationally expensive process. Additionally, this process iteratively minimizes the cost function with many forward runs and, as such, can not be parallelized. The computation cost of 4D-var is independent of the number of observations but can still be very large. It also requires storing many checkpoint files which can become very large for high spatial resolution and can have high storage costs. Gaussian Plume models are known for their simplicity and are often used for point source modeling (Bovensmann et al., 2010; Nassar et al., 2017; Wang et al., 2020). However, these models typically assume favourable conditions such as constant winds and flat topography, which may not always be the case.

Here, we focus on constructing the source-receptor relationship using a full-physics Lagrangian Particle Dispersion Model (LPDM). As such, this requires constructing $n$ footprints, i.e. one footprint for each measurement.

The LPDM can be used to construct a footprint by advecting an ensemble of particles backwards in time from the measurement sites using archived meteorology (e.g., Lin et al., 2003). These Lagrangian trajectories are agnostic to the choice of grid and can be easily mapped to a high spatial resolution. Additionally, for this study, $n$ is small as compared to $m$ and, therefore, it is computationally efficient to use the Lagrangian method of constructing $\mathbf{H}$.

The computation of footprints from a full-physics LPDM is an embarrassingly parallel problem and, as such, can easily be parallelized. However, this does not overcome the sheer volume of data that would need to be generated for a high-resolution GHG flux inversion. The construction of footprints can quickly become computationally intractable as the number of measurements increases, as is the case with dense observing systems.

In addition to the computational cost of constructing footprints described above, there is a storage cost associated with these footprints that increases with the number of measurements. This can also become burdensome as the number of observations increases. As an example, Turner et al. (2018) examined point sources in the Barnett Shale region in Texas and found that generating one week of footprints for hourly measurements made by a geostationary satellite instrument required more than 15 million simulations to construct the footprints and over 4 Tb to store them. This region represents less than 1% of the United States. As such, many previous studies investigating point sources have focused on a subset of cases with favourable atmospheric conditions, allowing them to utilize a simplified representation of atmospheric transport. For example, recent work estimated $CO_2$ emissions from individual power plants using a Gaussian plume (e.g., Nassar et al., 2017, 2021; Guo et al., 2023), which is only valid for steady winds.

Recent works have explored the potential of machine learning and other analytical methods to address computational bottlenecks while working with LPDMs. For example, Roten et al. (2021) developed an interpolation method using nonlinear weighted averaging to compute footprints. Cartwright et al. (2023) developed a convolution-based variational autoencoder which predicts footprints using a spatiotemporal Gaussian emulator. Brecht et al. (2023) used neural networks for super-resolution to improve FLEXPART trajectory calculations. Fillola et al. (2023) developed an emulator using gradient-boosted regression trees to predict the influence for each grid, one at a time. These studies provide a proof-of-concept for emulating full physics LPDM; however, they still require running the full physics LPDM simulations a significant number of times to conduct flux inversions. Fillola et al. (2023) developed a stand-alone emulator which can predict the near-field of the footprints at $\sim 35 \times 23$ km$^2$ resolution, however, they had to use LPDM simulations for the far field while using the emulator in an

inversion framework. Hence, these proof-of-concept studies are not entirely independent of LPDM simulations in an inversion framework. Additionally, their spatial resolutions are very coarse which is not ideal for high-resolution emission inventories.

Here we use FootNet, a computationally efficient deep-learning model which emulates the atmospheric transport (He et al., in press) to conduct GHG flux inversion at the kilometer scale. FootNet is a deep-learning model based on a U-Net architecture and trained on outputs from the Stochastic Time-Inverted Lagrangian Transport model (STILT; Lin et al., 2003). FootNet computes footprints at $1 \times 1$ km$^2$ resolution and is independent of the parent LPDM after the training process. As such, FootNet can be readily used in GHG flux inversions and, once trained, does not require STILT model outputs to predict the footprint. Here we evaluate the performance of FootNet in GHG flux inversion using the San Francisco Bay Area in Northern California as a case study.

## 2   Impacts of COVID-19 regulations on urban CO$_2$ fluxes in the San Francisco Bay Area as a case study

This study focuses on the San Francisco Bay Area in Northern California, we use hourly atmospheric CO$_2$ measurements from the Berkeley Environmental Air-quality and CO$_2$ Network (BEACO$_2$N; Shusterman et al., 2016; Turner et al., 2016; Shusterman et al., 2018; Turner et al., 2020; Asimow et al., 2024). BEACO$_2$N is a dense urban monitoring network with monitoring sites spaced $\sim$2 kilometers apart. This study uses hourly CO$_2$ measurements for the period of 02 February 2020 to 02 May 2020. Measurements are from approximately 35 BEACO$_2$N sites. These observations were used in a recent study from Turner et al. (2020) who evaluated the impact of COVID-19 restrictions on urban CO$_2$ fluxes. This high-resolution GHG flux inversion from Turner et al. (2020) will serve as a reference study to evaluate the performance of flux inversions using an ML model as a surrogate for the full atmospheric transport model.

As in Turner et al. (2020) and He et al. (in press), we use meteorological fields from the High-Resolution Rapid Refresh (HRRR) model at $3 \times 3$ km$^2$ spatial resolution to drive the STILT LPDM. Footprints from the STILT LPDM are then used as training data for FootNet (see He et al., in press, for details). The training data is sampled from 2018 to 2019, as such the timeline of this case study is not involved in the training data. FootNet predicts footprints at $1 \times 1$ km$^2$ resolution over a $400 \times 400$ km$^2$ spatial region centered on the San Francisco Bay Area. The setup for the CO$_2$ flux inversion is the same as in Turner et al. (2020). As such, we can compare the performance of the GHG flux inversion from FootNet with previously published work. Differences from the setup in Turner et al. (2020) will be emphasized in the text that follows.

## 3   Inferring CO$_2$ fluxes at high spatio-temporal resolution from atmospheric observations

Our goal in this work is to infer the fluxes of CO$_2$ using atmospheric observations. Bayesian inference is commonly used when estimating CO$_2$ fluxes using atmospheric observations. Building on the framework described in Section 1, we use Bayesian inference to relate the Probability Density Function (PDF) of the posterior ($P(x|y)$) to the observation likelihood PDF ($P(y|x)$) and prior PDF ($P(x)$) as follows:

$$P(x|y) \propto P(y|x)P(x). \tag{3}$$

Assuming a normal distribution for both $P(y|x)$ and $P(x)$ yields a closed form solution for the posterior distribution:

$$P(x|y) \propto \exp \left[ -\frac{1}{2}(y - Hx)^T R^{-1}(y - Hx) - \frac{1}{2}(x - x_a)^T B^{-1}(x - x_a) \right] \tag{4}$$

where, $x_a$ is the prior, $R$ is the observational error covariance matrix, $B$ is the prior error covariance matrix. The maximum a posteriori probability can be obtained by finding the minimum of the negative log-likelihood term within the exponential. The resultant cost function is:

$$\mathcal{J}(x) = \frac{1}{2}(y - Hx)^T R^{-1}(y - Hx) + \frac{1}{2}(x - x_a)^T B^{-1}(x - x_a). \tag{5}$$

Minimizing the cost function provides a closed-form estimate for the posterior fluxes ($\hat{x}$):

$$\hat{x} = x_a + (HB)^T (HBH^T + R)^{-1}(y - Hx_a) \tag{6}$$

where $\hat{x}$ is the posterior fluxes. It is important to note here that $B$ is an $m \times m$ matrix that is often computationally intractable for large state vectors. In this case study, $m$ is larger than 15 million and $B$ is computationally intractable. We use a Kroenecker product to decompose $B$ into temporal and spatial submatrices. Yadav and Michalak (2013) proposed a computationally efficient algorithm for serial computation of $HB$ and $HBH^T$ using this Kroenecker product without explicitly forming $B$. We further reduce the computation time of $HB$ computation by updating their algorithm to implement parallel computation (see Appendix A).

## 4   Relating observations to surface fluxes using footprints

This study uses footprints generated by both STILT and FootNet to relate surface fluxes to observations. Both of these models are driven by the same parent HRRR meteorology. STILT is a Lagrangian model built on the top of the Hybrid Single-Particle Lagrangian Integrated Trajectory (HYSPLIT) model and can produce either time-resolved (e.g. hourly) footprints or time-integrated footprints (i.e., the temporal dimension has been summed, yielding a 2-D matrix). STILT footprints generated for this study are hourly, going 72 hours backwards in time or until the trajectories leave the spatial domain (whichever is smaller). This is the same setup that was used in Turner et al. (2020). The FootNet model computes time-integrated footprints because it was deemed computationally infeasible to emulate the time-resolved footprints. Therefore, we need to devise an approach to allocate the FootNet footprints backwards in time. Here we use exponentially decaying weights to allocate the footprint across all 72 backhours. These weights are normalized such that the summation of the weights adds up to one and conserves the total magnitude of the footprint. This section compares the temporal allocation using exponential decay on time-integrated STILT footprints with the hourly time-resolved STILT footprints in a GHG flux inversion to investigate the additional error induced before using it with FootNet footprints in section 5.

Figure 1 shows the difference between hourly time-resolved footprints and the time-integrated footprints with exponentially decayed weights. The time-resolved footprints have a smaller sensitivity "blob" that moves as we go backward in time. This indicates that for any given time step, only the emission sources in the small shaded region influence our observation. This

highly time-resolved method assumes that the numerical schemes used for the transport and advection are highly accurate. In contrast, the time-integrated footprint with exponential decayed weights assumes a time-invariant spatial structure with decreasing magnitude at previous time steps. This plume decays with time such that time steps close to the time of observation have higher weights as compared to the plume 72 hours before the time of observation, which has negligible influence. This temporal allocation of the footprint will likely induce additional error.

We assess this error using a pair of flux inversions: 1) conduct a GHG flux inversion with time-resolved STILT footprints and 2) conduct a second GHG flux inversion using STILT footprints where the footprints have been time-integrated and temporally allocated as described above. This pair of GHG flux inversions will isolate the impact of this temporal allocation. Please note that all other inversion parameters are the same between these tests.

Figure 2 shows the results of these two GHG flux inversions in the Bay Area. It is important to emphasize that both of the inversions use the same set of footprints derived from the STILT model. The only difference is that one set of STILT footprints has been time-integrated and then reallocated temporally using exponential decay. The scatter plots compare observed $CO_2$ concentrations with the simulated $CO_2$ concentrations using prior and posterior fluxes on a validation dataset that was not used in the flux inversion. We use the same seed such that same validation concentrations are sampled in both cases. Overall, the posterior emission fluxes are in agreement. Interestingly, the second case where the footprints have been temporally reallocated do a better job at simulating independent observations with both the prior fluxes and the posterior fluxes. This can be seen in both the correlation and the mean squared error. Upon close examination of the scatter plots, it can be seen that the time-resolved scatter plots have a cluster of simulated concentrations around 410 ppm ($\sim$ background signal) even though actual concentrations are higher than that. This pattern is visible in the $CO_2$ concentrations simulated using both prior and posterior fluxes. On the other hand, this pattern is not visible in the $CO_2$ concentrations simulated using the posterior fluxes of the exponential decay case.

The poor performance of the time-resolved footprints is likely driven by transport errors in the STILT simulations. The time-resolved footprints are a more realistic representation of the source-receptor relationship, but not necessarily more accurate. The simulated transport could have large biases that will propagate into time-resolved footprints. For example, errors in the windspeed could lead to parcels advecting too fast and attributing fluxes to the wrong spatial location. In these flux inversions, the ocean and the SF Bay are assumed to be neither sources nor sinks. It seems that the sensitivity blobs in the time-resolved footprints are quickly advected over the ocean and hence the simulated concentrations are closer to the background signal even though the observed concentrations are higher. As such, the time-integrated footprints may be mitigating these transport errors and allowing GHG fluxes to be attributed correctly. Additionally, this pair of GHG flux inversions indicates that the exponential decay-based temporal allocation of footprints should not induce significant errors in the GHG flux inversion. All of the GHG flux inversion experiments that follow (i.e. Section 5) will use FootNet footprints and this temporal allocation of the footprints.

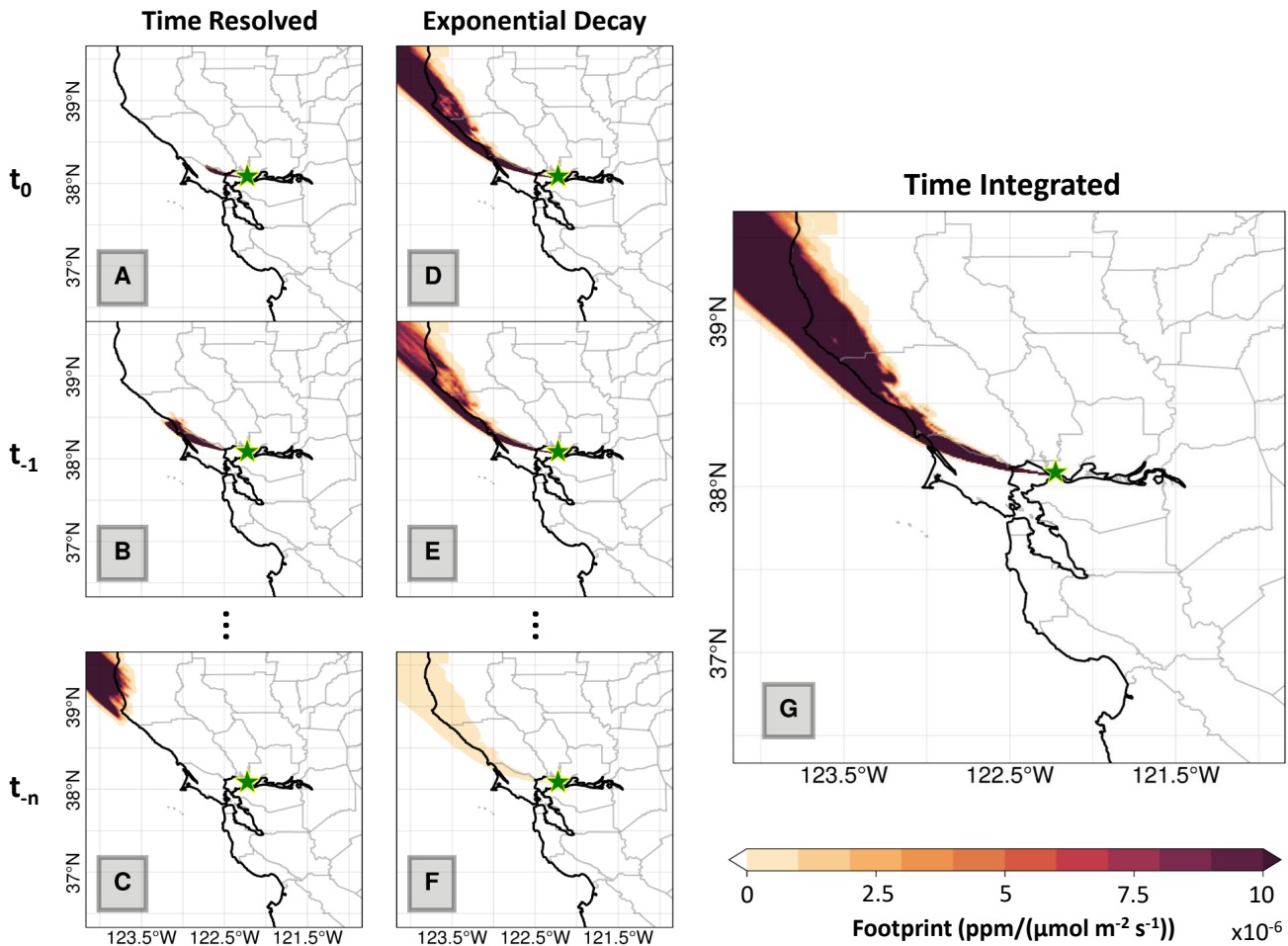

**Figure 1.** Two methods of representing temporal patterns in footprints for a single receptor. Green stars indicate the receptor location for the footprint. Panels (a–c) are time-resolved footprints computed directly from STILT. Panels (d–f) are time-integrated STILT footprints with the pattern allocated temporally using an exponential decay. Both representations have the same time-integrated spatial pattern (panel g).

## 5   GHG flux inversions with a machine learning surrogate model

Using the temporal allocation strategy described above, we can now assess the performance of an ML-surrogate transport model within a GHG flux inversion. Figure 3 shows conceptually how this ML model for atmospheric transport can be used within a GHG flux inversion. Fig. 3a shows the conventional GHG inversion framework using an LPDM. This framework begins with obtaining meteorological data for the spatio-temporal region of interest. This meteorological data is used to drive the full physics LPDM and, in turn, construct the footprints for all of the observations. Construction of these footprints is

computationally expensive and the footprints are typically archived prior to conducting the GHG flux inversion. Archiving

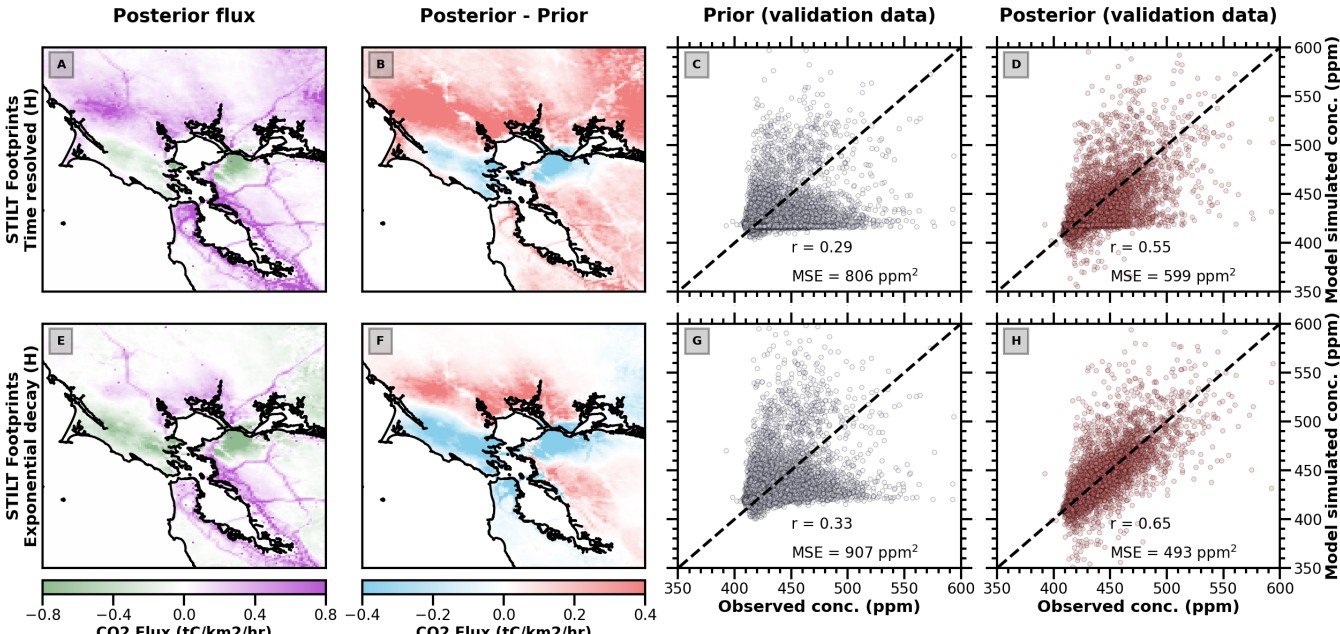

**Figure 2.** Comparison between $CO_2$ fluxes derived using time-resolved and time-integrated footprints with exponential decay for the STILT model. (Top row) Flux inversion results using time-resolved footprints from STILT. (Bottom row) Flux inversion results using time-integrated footprints with an exponential decay. (a and e) Posterior fluxes averaged over the study period from 2 Feb 2020 to 2 May 2020. Panels (b and f) show the difference between the posterior and prior fluxes. Panels (c and g) show a comparison of the model simulated concentrations using the prior fluxes against independent observations withheld from the flux inversion. Panels (d and h) are the same as (c and g), but using the posterior fluxes.

these footprints can have large data storage requirements for dense observations at high spatio-temporal resolution. Following the construction and archival of the footprints, researchers can then estimate GHG fluxes via Bayesian inference as described in Section 3.

Figure 3b shows the process to estimate GHG fluxes using an ML-model as a surrogate for the full-physics atmospheric transport model. The initial and final steps of the process are identical to the process described above. The difference is in the construction and archival of the footprints. We detail two approaches: 1) construct and archive the footprints or 2) construct the footprints "on-the-fly" during the GHG flux inversion. The former approach is similar to the process using the full-physics model, but the construction of the footprints uses the ML-surrogate model (e.g., FootNet). The latter approach of computing the footprints on-the-fly during the inversion is only feasible if the computation of the footprints can be done in near real time, otherwise additional computational expense would make the inversion prohibitively slow. The computation of a time-integrated footprint by the ML-surrogate model described by He et al. (in press) takes less than a second and, as such, may be sufficiently fast to facilitate computation of footprints on-the-fly as they are needed in the GHG flux inversion. In the following sections we

will evaluate the performance and the computational expense of GHG flux inversion frameworks using FootNet. These results will be compared to a GHG flux inversion using a full-physics atmospheric transport model (STILT).

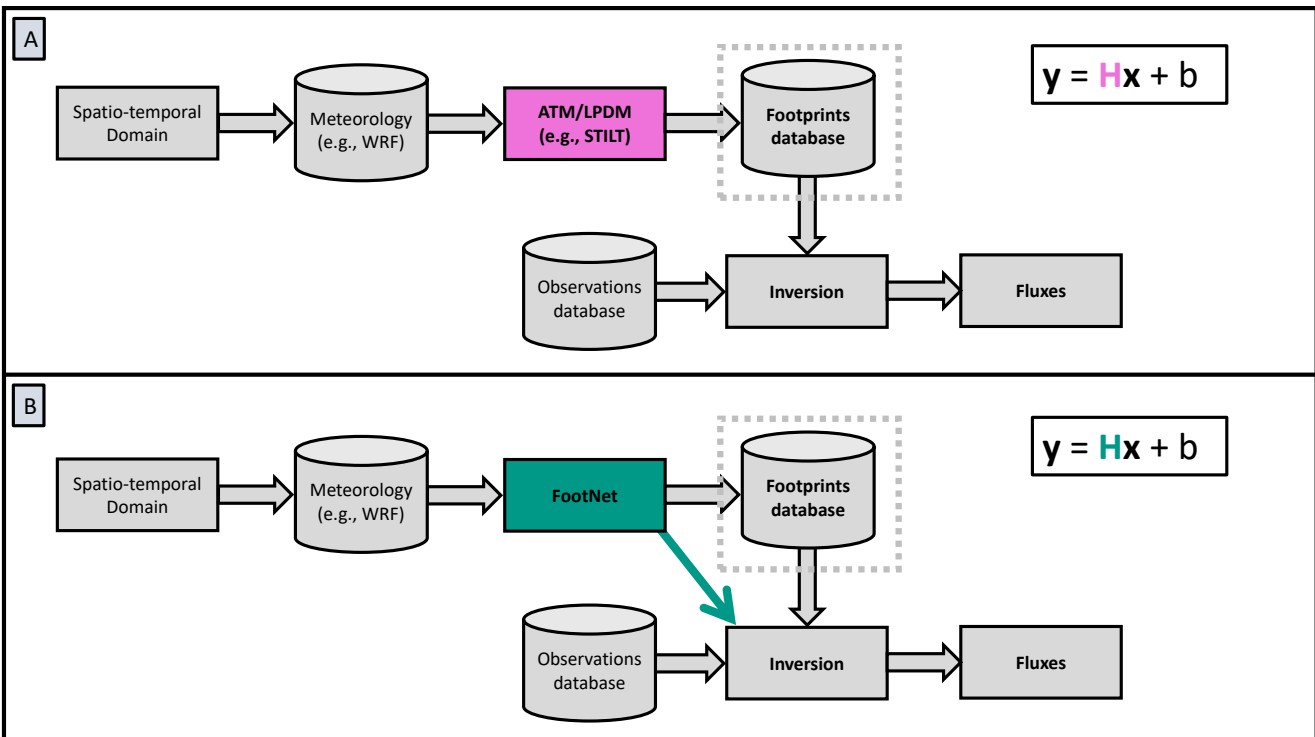

**Figure 3.** Conventional and FootNet-based GHG inversion framework. (Panel a) Conventional GHG flux inversion framework uses a full-physics LPDM simulation to compute footprints. (Panel b) Proposed GHG flux inversion framework using FootNet model to compute footprint. Arrow in panel b indicates how FootNet may be used to compute footprints "on-the-fly" as they are needed within the GHG flux inversion, bypassing the data storage step.

## 5.1 GHG flux inversion with FootNet v1

Figure 4 shows the results of a flux inversion using FootNet v1, the model described in He et al. (in press), with the flux inversion setup from Turner et al. (2020) who evaluated the impact of COVID-19 regulations on urban $CO_2$ fluxes in the San Francisco Bay Area. Briefly, we conduct hourly flux inversions at 1-km spatial resolution for overlapping 96-h windows. Each 96-hour window includes a 36 hour buffer on the 24-h period of interest. The prior error covariance matrix is decomposed using a Kronecker product (e.g., Yadav and Michalak, 2013). Upwind concentrations are taken from NOAA observations in the Pacific and AmeriFlux measurements in the Sacramento Delta.

From Fig. 4c and 4d, we observe that the posterior fluxes inferred using FootNet v1 perform substantially better than the prior fluxes when compared against independent validation data. Fig. 4a shows changes in physically meaningful locations

such as freeways in the San Francisco Bay Area as well as regions dominated by the biosphere. However, from panel b, we can see that these posterior fluxes do not appear to be in agreement with the fluxes inferred from Turner et al. (2020) using the full-physics model. FootNet v1 finds substantially higher fluxes throughout the domain than the inversion using the full-physics model. This suggests that FootNet v1 simulates realistic spatial patterns well but may be generating weaker footprints than STILT and may not be appropriately scaled. Additionally, the large differences in the far-field seen in Fig. 4b indicate that there could be an imbalance between the near-field and far-field of the footprints from FootNet v1.

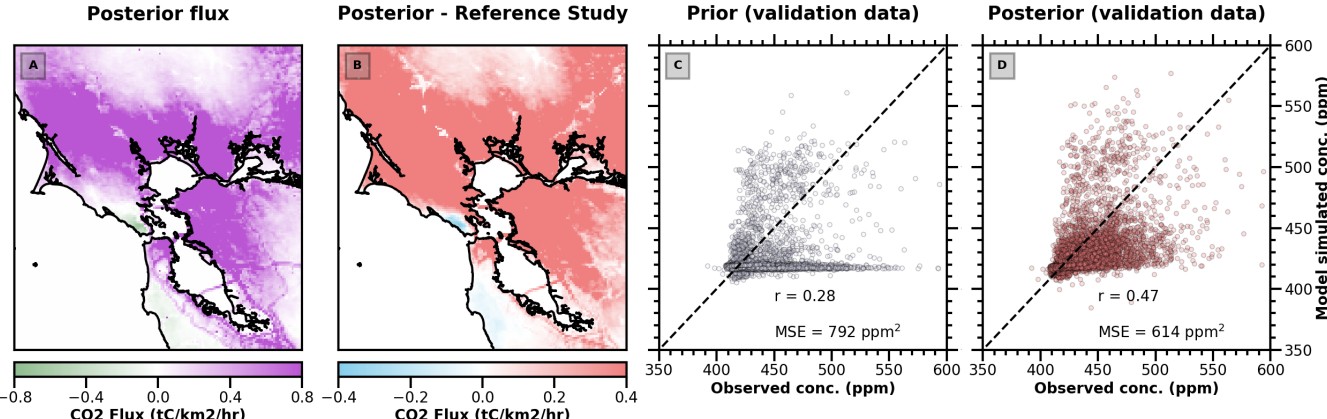

**Figure 4.** Urban $CO_2$ flux inversion results in the San Francisco Bay Area using FootNet v1 (He et al., in press). (Panel a) Posterior $CO_2$ fluxes averaged over the study period. (Panel b) Difference between the posterior fluxes inferred with FootNet v1 and fluxes inferred with a full-physics model (STILT). (Panel c) Comparison of $CO_2$ concentrations simulated with FootNet v1 and the prior fluxes against independent observations. (Panel d) Same as panel c but for posterior fluxes inferred from using FootNet v1. Reference study is Turner et al. (2020).

Figure 5 shows the cumulative influence of footprints for the full-physics model (STILT) and FootNet v1. These cumulative influence plots give an idea of the spatial regions that the observations are sensitive too. As alluded to above, there are strong similarities in the spatial patterns but FootNet v1 does indeed find a larger contribution from distant regions than STILT. This larger region of influence would result in FootNet allocating larger fluxes to distant regions than a flux inversion using STILT. This comparison of the regions of influence suggests that the FootNet model should be updated to improve the performance within a GHG flux inversion. We hypothesize two methods for improving the performance of FootNet: 1) changing parameters in the deep-learning architecture for FootNet and 2) adding input features to FootNet.

## 5.2 Updating the FootNet model to improve performance in flux inversions

As a first test, we train an additional variant of the FootNet model with an alternate formulation of the cost function (i.e., the cost function used to construct FootNet, not the cost function for a GHG flux inversion). This FootNet variant still relies on the underlying U-Net architecture. The FootNet v1 model used a mean-squared error (MSE; i.e., $\mathcal{L}_2$-norm) cost function in the gradient descent optimization. We hypothesize that an $\mathcal{L}_1$-norm cost function may improve the balance between the near-field

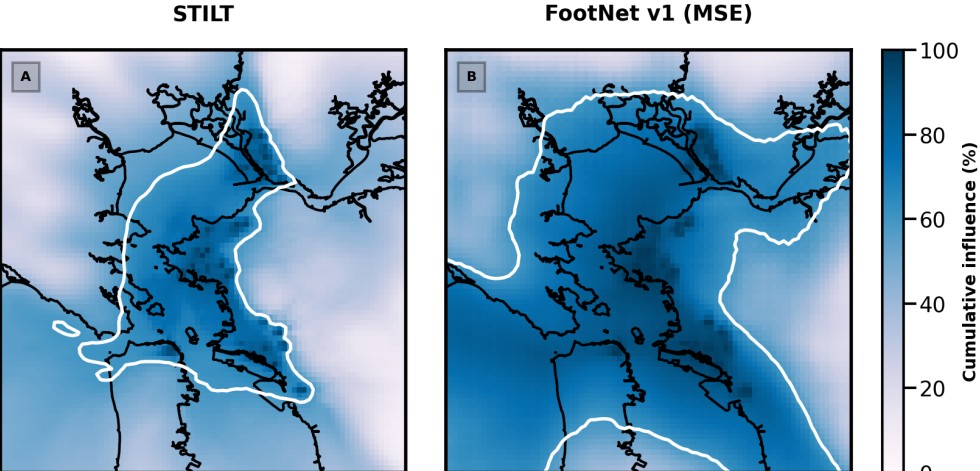

**Figure 5.** Cumulative region of influence for the BEACO$_2$N network in the San Francisco Bay Area. (Left column) Cumulative influence computed using STILT, a full-physics model. (Right column) Cumulative influence computed using FootNet v1 with an MSE cost function. White contour represents the region encapsulating the top 40% of the total influence on the BEACO$_2$N network (i.e., the region the observations are most sensitive to).

and the far-field. This is because the MSE cost function is sensitive to outliers, in contrast to an $\mathcal{L}_1$-norm that is less sensitive to outliers (cf., Bishop, 2006).

The top row of Figure 6 shows the cumulative influence for FootNet v1 using both an MSE and $\mathcal{L}_1$ norm cost function. Both
formulations of the cost function yield large spatial regions of influence. Based on this, we conclude that changing the cost function alone is insufficient to rectify the imbalance in the near-field and far-field footprints.

Two other FootNet model parameters we evaluate are the choice of activation function and the formulation of a log-transformation of the training data. The construction of FootNet v1 uses the Rectified Linear Unit (ReLU) activation functions to introduce non-linearity in the deep learning model architecture (He et al., in press). We assess the performance of Parametric
Rectified Linear Unit (PReLU) activation functions. These PReLU activation functions have parameters that can be tuned during the training process, giving FootNet additional degrees of freedom. FootNet v1 also uses a logarithmic transformation of the training data to help identify large-scale spatial patterns. We modify the logarithmic transformation to add a small number and ensure positivity as follows:

$$\Gamma(\mathbf{x}^*) = \log(\mathbf{x} + \epsilon) - \log(\epsilon) \tag{7}$$

where $\mathbf{x}$ is a real number and $\epsilon$ is a small value ($\epsilon = 10^{-3}$). Both of these updated parameter choices improved the performance of FootNet but did not fully rectify the near-field and far-field imbalance.

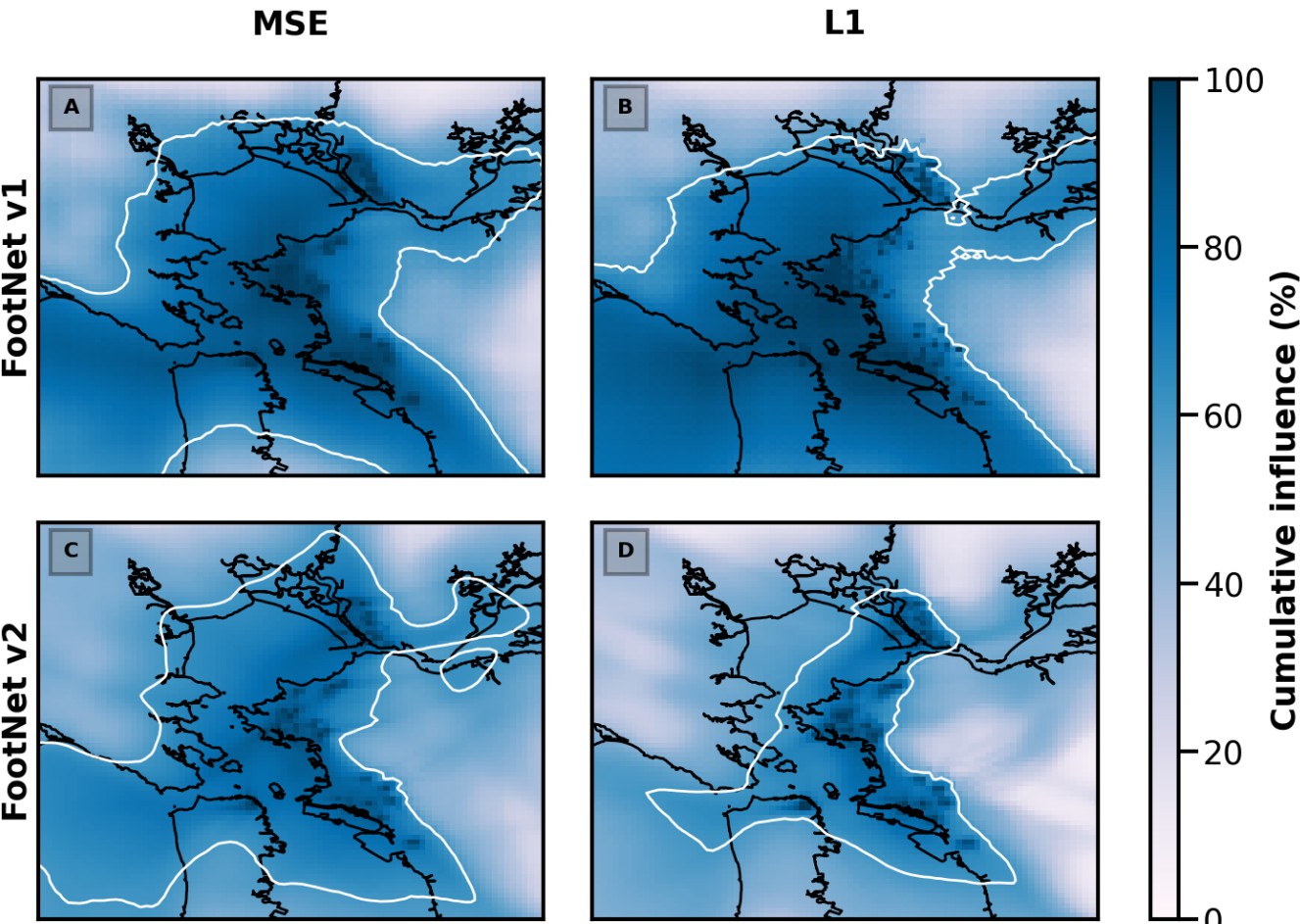

**Figure 6.** Same as Fig. 5 but for different variants of FootNet. (Top row) Cumulative influence for FootNet v1. (Bottom row) Cumulative influence for FootNet v2. (Left column) Using MSE cost functions. (Right column) Using $\mathcal{L}_1$-norm cost functions.

Finally, we trained 2 more FootNet models using additional input features shown in Table 1. We define these models with additional input features as "FootNet v2". FootNet v1 only uses meteorological data at the time of measurement and 6 hours prior. Here, we add meteorological data 12 hours back in time. This meteorological data include 10m zonal and meridional

winds, surface pressure, and planetary boundary layer (PBL) height. Winds are important for the model to learn advection, and surface pressure also acts as a proxy for the region's topography. We also use PBL height as it is an important input for computing footprints from the trajectories of the particles in a full physics LPDM (e.g. STILT; Lin et al., 2003). The choice of 12 hours is because many trajectories have not yet left the domain within 6 hours and, as such, meteorological information from 12 hours before the observation time may be important in constructing the footprint and assessing fluxes. We also include

distance from the receptor (both linear distance and an exponentially decaying distance). Distance from the receptor may help the model learn the optimal decay rate for the spatial pattern of the footprint (i.e., help the imbalance of the near-field and

**Table 1.** Input features for FootNet.

| Features | FootNet v1 | FootNet v2 |
|---|:---:|:---:|
| U10M (m/s) | ✓ | ✓ |
| V10M (m/s) | ✓ | ✓ |
| PBL Height (m) | ✓ | ✓ |
| Surface pressure (hPa) | ✓ | ✓ |
| Gaussian Plume at the time of measurement | ✓ | ✓ |
| Gaussian Plume 6h before measurement | ✓ | |
| Meteorology (U10, V10, PBL, Sfc Pres.) at the time of measurement | ✓ | ✓ |
| Meteorology (U10, V10, PBL, Sfc Pres.) 6h before measurement | ✓ | ✓ |
| Meteorology (U10, V10, PBL, Sfc Pres.) 12h before measurement | | ✓ |
| Distance from the receptor (m) | | ✓ |
| Combined Gaussian plumes network (mask) | | ✓ |

far-field). Finally, we include a spatial mask inferred from a network of Gaussian plumes. The spatial mask is based on winds at the time of measurement, 6 hours, 12 hours, 18 hours and 24 hours back in time and may help identify the important regions influencing our measurement.

Figure 6 shows the cumulative influence plots for FootNet v2 using both MSE and $\mathcal{L}_1$-norm cost functions. The $60^{\text{th}}$ percentile contours show a stronger resemblance to that of the full-physics STILT model (see Fig. 5a). The balance between the near-field and far-field is more in line with the cumulative influence inferred by the STILT model. As hypothesized, the model with the MSE cost function has a larger region of influence than the model using the $\mathcal{L}_1$-norm cost function. This allows the model to optimize the spatial decay structure of the footprints as it moves radially outward from the receptor location.

Notably, the MSE-based cost function indicates a larger sensitivity over the ocean as compared to the STILT footprint and the $\mathcal{L}_1$-norm. This is likely due to FootNet simulating "smoother" footprints than STILT (STILT exhibits sharp gradients at the edge of the footprint). The FootNet v2 model using an $\mathcal{L}_1$-norm cost exhibits a smaller region of influence than the STILT model.

    We can now assess the performance of the four models shown in Fig. 6 in the context of realistic GHG flux inversions. In

all cases, the models will be compared against the results of GHG flux inversion using the full-physics STILT model (Turner et al., 2020) and they will be evaluated against validation data from $CO_2$ observations withheld from the flux inversion.

    Figure 7 shows the comparison of these models against Turner et al. (2020) and the validation data (note, one of the cases is shown above in Fig. 4). As shown in Fig. 6, the near-field and far-field imbalance persists in FootNet v1 using an $\mathcal{L}_1$-norm cost function. Both variants of FootNet v2 show marked improvement in the near-field and far-field balance. Additionally, both

variants of FootNet v2 perform substantially better against independent validation data (see right column). Between the two variants of FootNet v2, we find that the MSE-based cost function performs best. This conclusion is based on the performance against independent validation data and comparison to the posterior fluxes from Turner et al. (2020). The comparison against

independent validation shows a strong correlation ($r = 0.68$) and no systematic biases in the residuals. The mean squared error against independent validation data is the lowest of the FootNet models tested ($\text{MSE} = 405$ $\text{ppm}^2$). FootNet v2 with an
MSE cost function also shows the smallest deviations from the posterior fluxes inferred from Turner et al. (2020), who used a computationally expensive full-physics model to relate the fluxes to observations. As such, we select FootNet v2 with an MSE cost function as the final model.

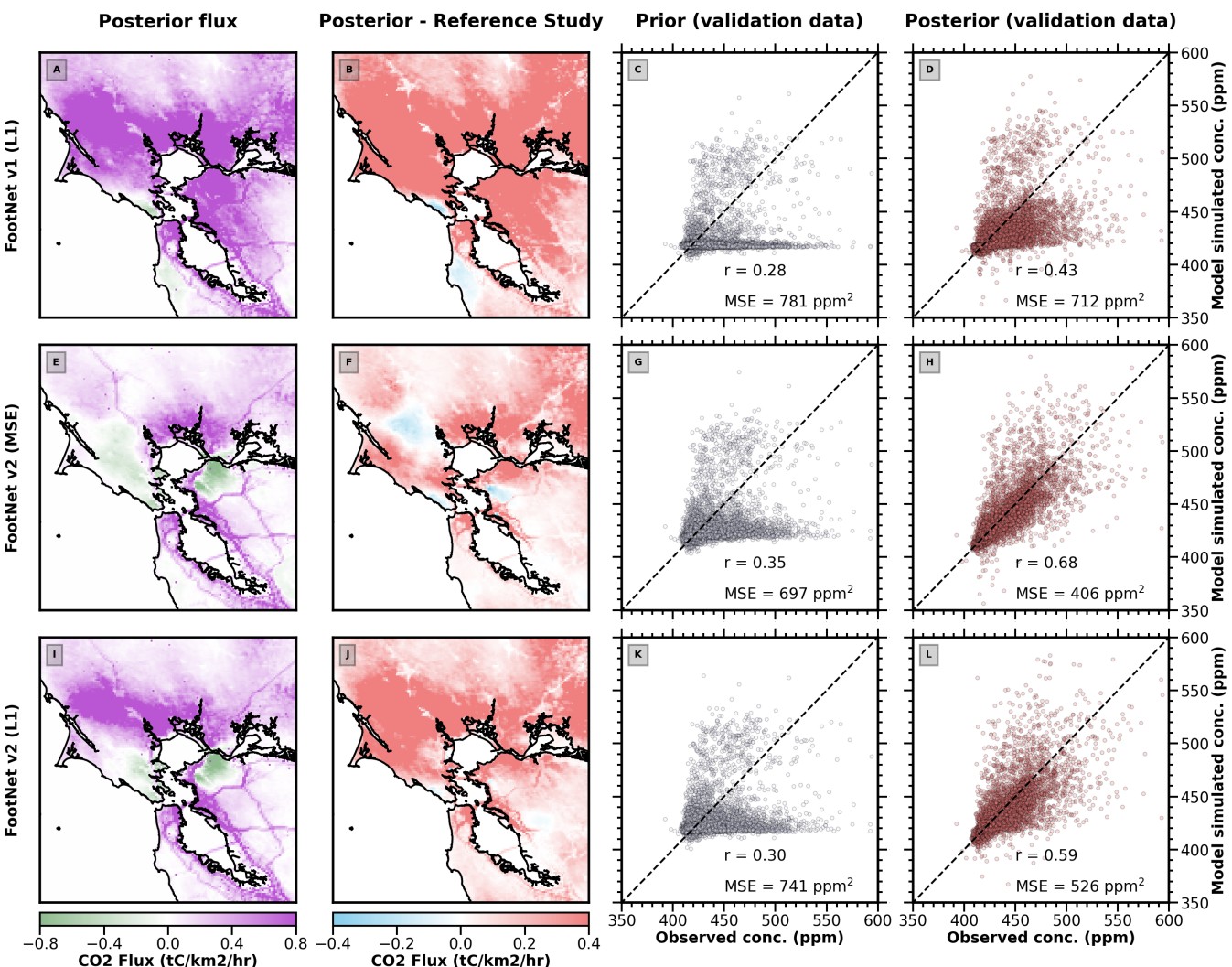

**Figure 7.** . Same as Fig. 5 but for three other variants of FootNet: FootNet v1 trained with an $\mathcal{L}_1$-norm cost function (top row), FootNet v2 trained with an MSE cost function (middle row), and FootNet v2 trained with an $\mathcal{L}_1$-norm cost function (bottom row). Reference study is Turner et al. (2020).

Figure 8 shows a direct comparison of FootNet v2 using an MSE cost function against the posterior fluxes inferred from Turner et al. (2020) using a full-physics model (STILT). The comparison shows the average fluxes for 6 weeks before COVID-

19 shelter-in-place, 6 weeks during shelter-in-place, the difference, and a comparison against independent validation data. Figs. 8a and 8e show a strong similarity in their $CO_2$ fluxes before the COVID-19 shelters were in place. We note some disagreements in the far-field, such as the northern and eastern parts of the domain. Figs. 8b and 8f correspond to the period during the shelter-in-place measures that decreased anthropogenic emissions. Both sets of posterior fluxes are in agreement during this period with clearly visible reductions in emissions from freeways and anthropogenic sources. The only notable

difference is near Tomales Bay and Point Reyes in the western portion of the domain. Figs. 8c and 8g show the difference in the $CO_2$ fluxes between these two periods. Both inversions largely agree in the Bay Area where the observations have the largest influence. Some disagreements can be seen to the east of Tomales Bay and in the Sacramento Delta. Finally, the right column of Fig. 8 shows the comparison against independent validation data for both flux inversions. Interestingly, we observe the FootNet v2 posterior fluxes to perform *better* than the posterior fluxes inferred using STILT. This is seen in both the

correlation coefficient ($r = 0.68$ for FootNet and $r = 0.65$ for STILT) and mean squared error ($\text{MSE} = 405$ ppm$^2$ for FootNet and $\text{MSE} = 493$ ppm$^2$ for STILT). This begs the question, *"why would a machine learning surrogate model perform better than the full-physics model in a GHG flux inversion?"*

FootNet was designed to emulate the STILT model (a full-physics atmospheric transport model), as such it is surprising to see the ML surrogate model outperform the full-physics model it was trained on when used in a GHG flux inversion. The

explanation for this paradox is that while STILT is a more realistic representation of the transport it is not necessarily more accurate. ML models often give predictions that tend towards the mean. In the context of atmospheric transport and footprints, this results in a FootNet simulating a smoother and more diffuse spatial pattern than STILT (He et al., in press). When used in a GHG flux inversion, the sharp gradients simulated with STILT mean that small errors in windspeed or direction could lead to fluxes being allocated to the incorrect spatial location. In the context of the GHG flux inversion, this diffuse spatial pattern

simulated by FootNet can potentially mitigate transport errors in the flux inversion. An important takeaway from this work is that using an ML-surrogate model in a flux inversion can *potentially* outperform the computationally expensive full-physics model. However, this result is unlikely to be universally true as there will be cases where smoother spatial patterns induce errors in the flux inversion (e.g. when the true trajectories are localized). The performance of the ML-surrogate model will almost certainly vary on a region-by-region basis and additional tests are needed to assess the extent of this finding.

**6  Computational cost of the GHG flux inversions**

Table 2 shows the computational and storage cost analysis for both STILT and FootNet v2 for the 3-month study period. The construction of footprints for all of the measurements using the full physics-based STILT model is computationally expensive. It takes roughly an hour (upper bound) to construct a single footprint using STILT. As such, it can take more than 8 years to construct all the footprints required for this study if one were to construct them sequentially. Parallel computation of the

footprints on a 32-core machine can reduce the time to 3 months. Using multiple nodes can further reduce the time to a few days. However, this reduction in time comes with an infrastructure cost. Given the computational expense in generating these footprints, researchers typically store these time-resolved footprints which can take ∼470 GB of space for this study period.

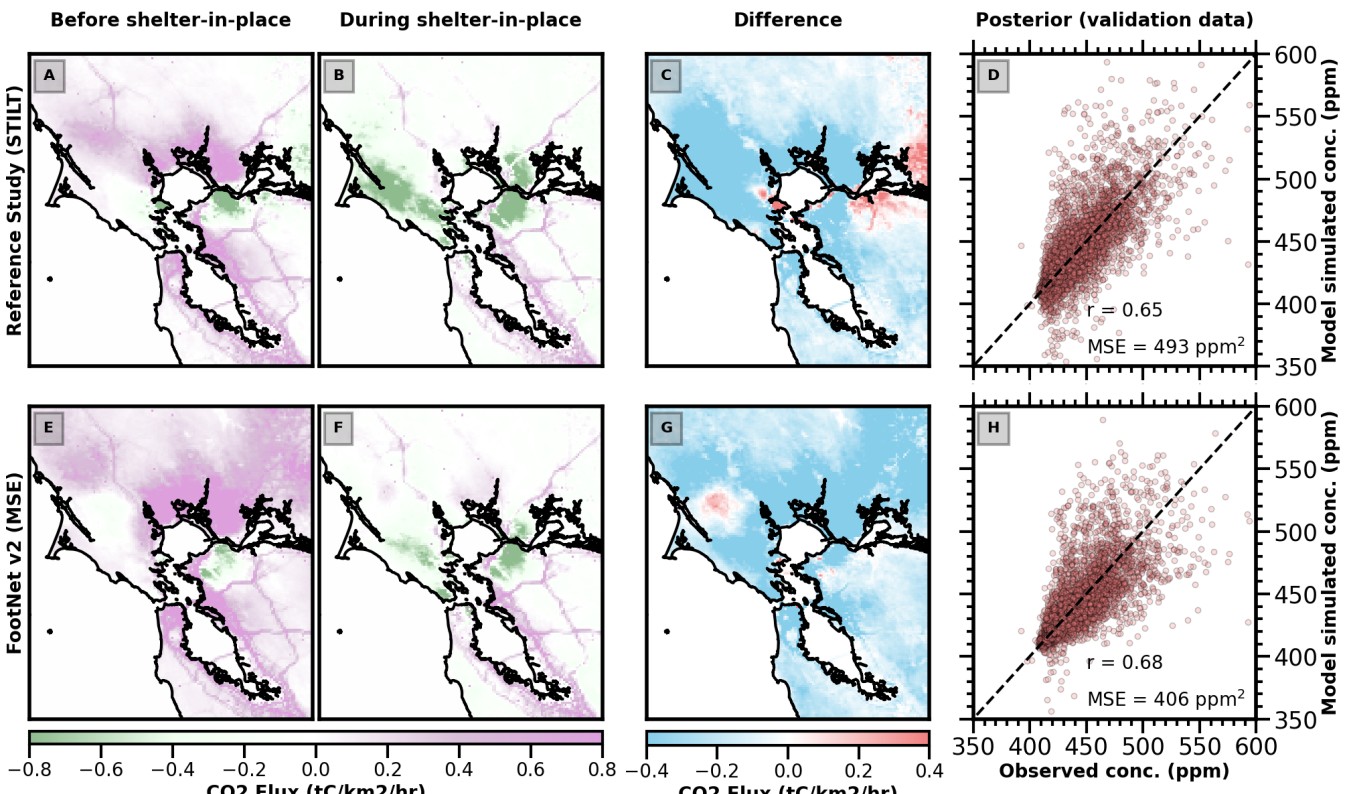

**Figure 8.** Urban $CO_2$ fluxes in the San Francisco Bay area inferred using atmospheric observations. (Top row) Posterior $CO_2$ fluxes inferred using footprints from STILT, a full-physics transport model. (Bottom row) Posterior $CO_2$ fluxes inferred using Footnet v2 with an MSE cost function. Panels (a and e) show the $CO_2$ fluxes averaged over 6 weeks before the COVID-19 shelter-in-place orders. Panels (b and f) show the $CO_2$ fluxes averaged over 6 weeks during the COVID-19 shelter-in-place order. Panels (c and g) show the difference between these time periods. Panels (d and h) compare predicted $CO_2$ concentrations using the posterior fluxes with independent $CO_2$ observations withheld from the flux inversion. Note: both inversions use time-integrated footprints that have been temporally allocated based on an exponentially decaying weight.

The computation of footprints using FootNet v2 is fast. It takes 3.5 hours to compute the same footprints using a single NVIDIA A2 GPU card on one core and 24 hours using a 32-core machine; this is a $650\times$ and $85\times$ speedup on similar hardware, respectively. This time includes reading input data, constructing footprints, and writing to disk. He et al. (in press) mention that it takes 0.08 seconds to compute a single footprint after loading input data. This speedy computation from FootNet allows for near real-time construction of footprints within the GHG flux inversion. In other words, one can compute footprints on-the-fly as they are needed, rather than storing the footprints on the disk. Additionally, it may allow researchers to conduct sensitivity runs on meteorological parameters (e.g., the PBL height) during the flux inversion by including meteorological

**Table 2.** Construction and storage cost to compute footprints with STILT and FootNet[a]

| | STILT | | | FootNet | | |
|---|---|---|---|---|---|---|
| | **Serial** | **Parallel** | **Parallel (multi-node)** | **CPU** | **GPU (archival[b])** | **GPU (on-the-fly[c])** |
| | 1 core[d] | 32 cores[d] | 320 cores[d] | 32 cores[d] | 1 GPU[e] | 1 GPU[e] |
| **Construction[f]** | 8.4 years | 3.15 months | 10 days | 1.1 days | 3.5 hours | up to 3 minutes[g] |
| **Storage** | 467 GB | 467 GB | 467 GB | 45 GB | 45 GB | 0 GB[h] |

[a]Computation times may change slightly based on factors such as hardware usage, resource availability etc.

[b]Flux inversion using archived footprints.

[c]Footprints computed as they are needed during the flux inversion.

[d]Intel(R) Xeon(R) Gold 6226R CPU @ 2.90GHz.

[e]NVIDIA A2 GPU, 16 GB GDDR6.

[f]Constructing 73,703 footprints.

[g]~800 observations for a one-day inversion with a 96-hour window.

[h]No archival of footprints in the on-the-fly framework.

parameters as a hyperparameter in the inversion. We note that the storage cost is lower for FootNet because STILT is saving time-resolved footprints.

The serial computation of **HB** matrix based on the computationally efficient algorithm proposed by Yadav and Michalak (2013) can also become computationally expensive. In this study, we implemented a parallel computation of **HB** matrix based on the algorithm detailed in Yadav and Michalak (2013). This parallel implementation is approximately 27 times faster than the serial implementation. Appendix A discusses the parallel implementation of **HB** matrix computation.

## 7 Conclusions

Near real time quantification of greenhouse gas (GHG) fluxes is important for monitoring, reporting, and verification of GHG fluxes to ensure climate goals are met. Here we demonstrate how machine learning (ML) models can be used as a surrogate for the full-physics atmospheric transport models in a GHG flux inversion, alleviating a computational bottleneck. This work updates the deep learning architecture of FootNet v1 (He et al., in press) to improve the performance in a GHG flux inversion. This updated deep learning model for atmospheric transport (FootNet v2) outperforms the full-physics model in an inversion estimating urban $CO_2$ fluxes at high spatio-temporal resolution in the San Francisco Bay Area. Further tests are required to investigate the generalizability of this finding.

A potential barrier to using FootNet within a GHG flux inversion is that FootNet computes a 2-D spatial pattern of time-integrated footprints whereas the full-physics model (STILT) generates time-resolved footprints. To overcome this, we temporally allocate the footprints using an exponentially decaying weight such that there is a time-invariant spatial structure with decreasing magnitude at previous time steps. Further, we compare predicted concentrations after conducting flux inversion us-

ing both the time-resolved and time-integrated footprints with temporal allocation. We observe that time-integrated footprints perform better as they can mitigate transport errors in the time-resolved representation. The time-resolved footprints are a more realistic representation of the source-receptor relationship, but not necessarily more accurate. Additional tests are needed to understand the transport errors in the time-resolved footprint and the broader applicability of the exponential decay footprints. This overcomes a potential barrier to using the ML-surrogate model in a flux inversion.

A preliminary flux inversion using FootNet v1 suggested there was a bias in the balance between the near-field and far-field footprints. We constructed additional variants of the FootNet model and evaluated them in an urban $CO_2$ flux inversion. Performance was evaluated against independent observations that were withheld from the flux inversion. Ultimately, we developed a new model (FootNet v2) that includes additional input features. We find that FootNet v2 outperforms the full-physics model in the flux inversion in this particular study. This is likely because the FootNet v2 footprints have a smoother spatial structure than the full-physics model. This smoother spatial structure can help mitigate transport errors. Additionally, this machine-learning surrogate model allows for a $650\times$ speedup in the construction of the footprints as compared to the full-physics model. This speedup allows for on-the-fly computation of footprints during the inversion, opposed to archiving footprints prior to the GHG flux inversion.

Previous work has shown that the distribution of GHG sources may be skewed with a "heavy-tail" of super emitters. This suggests that the assumption of Gaussian distribution for the prior PDFs may not be accurate. Stochastic methods such as Markov Chain Monte Carlo can allow one to specify non-Gaussian prior PDFs as well as jointly solve for meteorology (e.g. uncertainties in PBL height). However, it is currently infeasible to implement with traditional models as it requires evaluation of the forward model many times, which is computationally intractable. FootNet can compute the footprints in near-real-time, making it feasible to use these methods to estimate posterior emissions. This can be one potential application of machine learning surrogates of atmospheric transport in improving the flux estimates.

Overall, FootNet alleviates a computational bottleneck when working with dense GHG observing systems, such as those from urban monitoring networks and next-generation satellite measurements (e.g, MethaneSat and Carbon-I). The computational efficiency of FootNet allows for near-real-time emission monitoring of GHGs along with other non-reactive trace gases. This work demonstrated the utility of FootNet in quantifying urban $CO_2$ fluxes in a case study and future work is needed to extend this framework to a larger region such as the contiguous US or total column measurements. Nevertheless, ML-surrogate models such as FootNet represent a promising direction for efficiently interpreting the growing volume of observational data from next generation observing systems.

*Code availability.* The code for this study is available at https://github.com/nd349/Bayesian and https://doi.org/10.5281/zenodo.13750963.

*Data availability.* $CO_2$ data is available at http://beacon.berkeley.edu/. The FootNet training data used in this study is same as He et al. (in press), The posterior fluxes are uploaded to https://doi.org/10.5281/zenodo.13750963.

## Appendix A: Parallel implementation of HB matrix multiplication

Here we use a shared memory parallelization technique to compute $\mathbf{HB}$ from $\mathbf{H}_{n \times pr}$, the temporal prior error covariance matrix ($\mathbf{D}_{p \times q}$), and the spatial prior error covariance matrix ($\mathbf{E}_{r \times t}$). This method is similar to the algorithm described in Yadav and Michalak (2013). The primary difference from Yadav and Michalak (2013) is we form $\mathbf{HB}$ on the shared memory and use a multi-threading approach to iterate over the $q$ columns of $\mathbf{D}$ simultaneously such that a thread performs the following operation on the $k^{\text{th}}$ column of $\mathbf{D}$:

1. Multiply each $(n \times r)$ block of $\mathbf{H}$ by the elements of the $k^{\text{th}}$ column of $\mathbf{D}$ and add these blocks.

    2. Multiply the resulting $n \times r$ matrix by $\mathbf{E}_{r \times t}$ to obtain the $k^{\text{th}}$ $n \times t$ column block of $\mathbf{HB}$ matrix

    3. Update the $k^{\text{th}}$ $n \times t$ column block of $\mathbf{HB}$ matrix

    4. End the thread operation

    This method is limited by the number of threads and the memory available for the matrix multiplication. The larger the
number of threads are faster the multiplication can be, provided that there is enough memory available for each thread.

*Author contributions.* N.D., T.L.H., and A.J.T. designed the research study; N.D. and T.L.H. trained the models, conducted flux inversions, and analyzed the results; N.D., T.L.H., and A.J.T. wrote the manuscript.

*Competing interests.* The authors declare that there is no conflict of interest.

*Acknowledgements.* This work is supported by a NASA FINESST Grant (80NSSC22K1557) to N.D. and a NASA Early Career Faculty
Grant (80NSSC21K1808) to A.J.T. and T.H. We acknowledge funding from the Environmental Defense Fund, whose work is supported by gifts from Signe Ostby, Scott Cook and Valhalla Foundation. This project is supported in part by Schmidt Sciences through the VESRI program. N.D. also acknowledges funding from the Integral Environmental Big Data Research Fund of the University of Washington. We thank the three anonymous reviewers for their thoughtful feedback and constructive comments. Their suggestions helped in improving the clarity and quality of this manuscript.

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
