# Peer review of "High-resolution greenhouse gas flux inversions using a machine learning surrogate model for atmospheric transport"

_EGUsphere, 2024_

## Author Comment (AC1)

We thank the reviewers for their time and constructive comments on our manuscript. Our responses are color-coded in blue.
* * *
To preface our response (and previously mentioned in our separate responses to Reviewer #1), this manuscript is part of a bigger project to develop a machine-learning emulator of atmospheric transport that can compute the footprint for a receptor in any region at any time and can be used in a flux inversion. This project consists of distinct projects:

1. He et al., GMD (in press) is a proof of concept to demonstrate that we can emulate footprints using machine learning for surface observations. This work focused on two regional case studies.
2. This manuscript under consideration here (Dadheech et al., under review) is a demonstration that machine learning emulators can be used in a GHG flux inversion with minimal error induced. This work focused on comparing against previously published work in a single region.
3. The final paper (in prep) aims to generalize this methodology to any region for both surface and column-averaged observations.

Below are our responses to the comments from reviewers:

**Reviewer #1**

See responses: https://egusphere.copernicus.org/preprints/2024/egusphere-2024-2918

**Reviewer #2**

Line 9-11 (abstract) Authors mention that, surprisingly, "the updated FootNet model out-performs the full-physics model when used in a flux inversion". However, they don't have a rational explanation for this and speculate as: "This improved performance is likely because atmospheric transport simulated with a full-physics transport model is not necessarily more accurate. The more simplistic representation of transport in the machine learning model helps to mitigate transport errors". Suggest to drop this speculative discussion, as it may happen that in the next version or a case study the ML and full-physics footprints will have other biases and advantage of the ML will be lost.

We thank the reviewer for this suggestion. It is presently unclear if this result is only present for the SF Bay Area region or is a more general result. We plan to revisit this in the third manuscript where we can more confidently say if this is just due to regional

variance or a general finding. **As such, we have dropped the explanation of FootNet's better performance from the abstract and will revisit this in the forthcoming analysis.**

Line 21-22 Giving the list of references here, authors implicitly limit the type of models useful for inverse modeling at high resolution to Lagrangian, while there are successful examples of using Eulerian models for the same purpose (eg Steiner et al, 2024). On the other hand, the possible applications of ML-based footprint simulators do not have to be regional as there are global models using Lagrangian footprints (eg Nayagam et al, 2024, Janardanan et al 2024) facing same or bigger computational challenges as for the regional ones.

We thank the reviewer for this comment. The reference list was indeed largely focused on Lagrangian approaches because our manuscript uses a Lagrangian emulator. We have included additional references to Eulerian models:

Line 20-23: However, the large computational and storage costs associated with full physics atmospheric transport models in the current inversion framework limit our ability to perform near-real-time emissions monitoring from urban to global scales (Roten et al., 2021; Varon et al., 2022a; Cartwright et al., 2023; Fillola et al., 2023; Nayagam et al., 2023; Steiner et al., 2024; Janardanan et al., 2024).

Lines 28-30 Can add satellite-based studies of point sources (eg Janardanan et al 2016).

We added a reference for satellite-based studies of point sources.

Line 28-31: Previous work has shown the importance of point sources for methane emissions (e.g., Brandt et al., 2014; Zavala-Araiza et al., 2015; Frankenberg et al., 2016; Duren et al., 2019; Lauvaux et al., 2022; Chen et al., 2022; Cusworth et al., 2022; Sherwin et al., 2023; He et al., 2024), and urban & localized sources for CO2 (e.g., Hutyra et al., 2014; Janardanan et al., 2016; Turner et al., 2020; Wu et al., 2020; Kiel et al., 2021).

Lines 57-58 Note that for large n, one may opt to using forward transport instead, either Lagrangian or Eulerian or plume-based like PMIF (Wang et al, 2020)

Excellent point.  We have updated the text.

Line 61-63: Gaussian Plume models are known for their simplicity and are often used for point source modeling (Bovensmann et al., 2010; Nassar et al., 2017; Wang et al., 2020}. However, these models typically assume favorable conditions such as constant winds and flat topography, which may not always be the case.

Line 224 Table 1. Looking from the experience of applying limited set of parameters for describing PBL mixing before 3-D dynamic models of turbulence (eg Hanna, 1984), the choice of driving variables does not look optimal. Why don't include surface stress, Monin-Obukhov length, 100 m or mid-PBL winds, for example?

We conducted a number of early experiments with larger sets of parameters but found that it did not improve the performance of the model.  The STILT model directly uses PBL height as a parameter to compute footprints (Lin et. al., 2003).  Given our previous experience with the STILT model, it was unsurprising that PBL height was found to be key for predicting footprints. In the end, we opted for a parsimonious model.  We have updated the text as follows:

Line 251-252: We also use PBL height as it is an important input for computing footprints from the trajectories of the particles in a full physics LPDM (e.g. STILT; Lin et. al., 2003).

Line 315-320 There is an impression that the ad hoc replacement of time-resolving footprints with a decay-based model will not be universally applicable, and it should be mentioned as a limitation of the proposed method.

This is a fair concern. We do think it may be universally applicable, but our study is currently limited to two regions, thus the caveat here. We have updated the text as follows:

Line 343-346: We observe that time-integrated footprints perform better as they can mitigate transport errors in the time-resolved representation. The time-resolved footprints are a more realistic representation of the source-receptor relationship but not necessarily more accurate. Additional tests are needed to understand the transport

errors in the time-resolved footprint and the broader applicability of the exponential decay footprints.

**Reviewer #3**

General Comment:

The manuscript *"High-Resolution GHG Flux Inversions Using a Machine Learning Surrogate Model for Atmospheric Transport"* presents an important study showcasing the potential of ML-based emulators for atmospheric transport in enabling fast computation of observation footprints used in flux inversions. The demonstrated gains in computational speed and storage compared to traditional methods relying on full-physics models are convincing and could pave the way for near-real-time GHG flux monitoring using dense observational systems, such as the new generation of satellite instruments. Provided the comments below are addressed, this paper would be suitable for publication in this journal.

One key claim of this work is that the proposed ML approach (FootNet v2) outperforms the full-physics model in the flux inversion. The authors attribute this to the smoother spatial structure of the FootNet v2 footprints, which they hypothesize helps mitigate transport error. Given the broad implications of this finding for the field, this important statement should be supported with more evidence than the statistical results obtained from a single case study. Two potential avenues for further substantiation are:

1.Extending the comparison to other cases reflecting different meteorological conditions.

2.Conducting an OSSE (Observing System Simulation Experiment).

I suggest the authors conduct at least an OSSE experiment, which would entail:

a) Generating synthetic observations from a reference "true" emission field.

b) Generating a prior ensemble of HRRR meteorology (or from another model) and fluxes.

c) Performing multiple inversions using both STILT and FootNet v2.

Each inversion would use a different meteorological realization to simulate transport error. The hypothesis presented in this study suggests that the ML surrogate, by smoothing the transport patterns, would produce better inversion results on average than STILT. This could be evaluated by comparing the standard deviations of inversion errors (relative to the known true fluxes) and the biases.

Although this approach is idealized, it would provide more robust statistical evidence to support the claim that the ML surrogate yields better performance than the full-physics model.

We thank the reviewer for this excellent suggestion. This study serves as a proof-of-concept that demonstrates the potential of using a machine-learning surrogate model for atmospheric transport in high-resolution flux inversion. The outperformance of FootNet carries significant implications for the field. We agree that conducting an OSSE study, as suggested, would provide more robust statistical evidence to support our results. However, even with such an OSSE we would be wary of claiming that this is universally true because our study is limited to one region. The OSSE proposed above would carry a large computational burden because we would need to generate footprints with another model (or wind fields). In light of this, we have updated the text to remove some of the conclusions that we don't feel are fully supported by the current simulations.

Additionally, we are in the process of acquiring meteorological data from the Global Forecast System (GFS) and plan to perform an OSSE similar to the one proposed by the reviewer using FootNet v3 that we are fine-tuning. This model is generalizable to all of CONUS and would allow us to make more general claims about the applicability of the findings to other flux inversions.

Line 336-338: This updated deep learning model for atmospheric transport (FootNet v2) outperforms the full-physics model in an inversion estimating urban $CO_2$ fluxes at high spatio-temporal resolution in the San Francisco Bay Area. Further tests are required to investigate the generalizability of this finding.

Line 351-352: We find that FootNet v2 outperforms the full-physics model in the flux inversion in this particular study.

Minor Comments:

Introduction:

The authors should discuss the use of variational methods as an alternative approach to address high-dimensional inversion problems involving large flux and/or observation spaces. In this framework, transport Jacobians do not need to be explicitly constructed, and efficient minimization algorithms enable rapid computation of mean posterior fluxes.

Thank you for the suggestion. We have added a discussion on the variational methods along with Lagrangian and Eulerian models in the manuscript:

Line 57-61: Variational methods such as 4D-var can be used with large state and observation space. However, it requires computing an adjoint, which is a computationally expensive process. Additionally, this process iteratively minimizes the cost function with many forward runs and, as such, can not be parallelized. The computation cost of 4D-var is independent of the number of observations but can still be very large. It also requires storing many checkpoint files which can become very large for high spatial resolution and can have high storage costs.

Conclusion:

It would be useful to discuss the potential application of ML surrogates for atmospheric transport in performing MCMC inversions. This approach could provide full posterior probability density functions (PDFs) without constraining the prior PDFs to specific forms, such as Gaussian priors.

This is a fantastic point. The ML surrogates for atmospheric transport can indeed be helpful in computing the posterior PDF using methods such as MCMC, which does not assume Gaussian distribution for the prior PDF. We have included a discussion in the conclusion section on this:

Line 357-363: Previous work has shown that the distribution of GHG sources may be skewed with a "heavy-tail" of super emitters. This suggests that the assumption of Gaussian distribution for the prior PDFs may not be accurate. Stochastic methods such as Markov Chain Monte Carlo can allow one to specify non-Gaussian prior PDFs as well as jointly solve for meteorology (e.g. uncertainties in PBL height). However, it is currently infeasible to implement with traditional models as it requires evaluation of the forward model many times, which is computationally intractable. FootNet can compute the footprints in near-real-time, making it feasible to use these methods to estimate posterior emissions. This can be one potential application of machine learning surrogates of atmospheric transport in improving the flux estimates.

---

## Author Response (AR2)

We thank the editor for the comment. Our response is color-coded in blue.

Dear authors, Thank you for your efforts in addressing the reviewer comments to the best of your abilities. Kindly acknowledge the contribution of the three anonymous reviewers in the 'Acknowledgements' section since their comments and the subsequent revisions made the manuscript better and helped clarify some of the finer points about FootNet for the readers. Once this is addressed, the manuscript can be published in ACP - your work presents a new paradigm for doing flux inversions, which while not without flaws, provides a new tool for the carbon cycle community while working with massive data volumes and deliver low latency flux estimates.

We thank the editor for this suggestion. We have added an acknowledgment of the contribution of the three anonymous reviewers who helped improve this manuscript.

Line 392-394: We thank the three anonymous reviewers for their thoughtful feedback and constructive comments. Their suggestions helped in improving the clarity and quality of this manuscript.